# Serological responses to killed oral cholera vaccine (OCV) when given 4 years after initial receipt of OCV in Cameroon: A randomized controlled trial

Jerome Ateudjieu[1,2,3]*, Ketina Hirma Tchio-Nighie[1,2], Etienne Guenou[1,2,4], Carrel Fokou[1], Collins Buh Nkum[1], Landry Beyala Bita'a[1], Winny Dora Ateudjieu-Kenfack[5,6], Sabine Nanfak[1], Amanda K. Debes[7], David A. Sack[7]

**1** Department of Health Research, M.A. SANTE, Yaounde, Cameroon, **2** Department of Public Health, Faculty of Medicine and Pharmaceutical Sciences, University of Dschang, Dschang, Cameroon, **3** Clinical Research Unit, Division of Health Operations Research, Cameroon Ministry of Public Health, Yaounde, Cameroon, **4** National Public Health Laboratory, Ministry of Public Health, Yaounde, Cameroon, **5** Department Microbiology, Faculty of Sciences University of Yaounde 1, Yaounde, Cameroon, **6** Center of Research on Emerging and Reemerging Diseases, Yaounde, Cameroon, **7** Johns Hopkins Bloomberg School of Public Health, Baltimore, Maryland, United States of America

* jateudj@masante-cam.org

## Abstract

Oral cholera vaccination (OCV) is one of strategies to help control cholera with two doses recommended as initial vaccination and booster doses recommended 3–5 years later to prevent future outbreaks. There is no actual recommendation on the number of doses required during booster vaccination. An OCV campaign conducted in Mogode-Cameroon provided an opportunity to determine if revaccination four years after would induce a booster response. 350 people living in Mogode-Cameroon, who received 0, 1 or 2 doses of OCV four years earlier, were stratified into age groups, < 5, 5–14, and >14 and randomized into two arms. Arm A received a single dose and Arm B received two doses 14 days apart. We aimed to determine if people who had received OCV four years earlier responded with a booster immune response compared to those not vaccinated. Serum samples were collected at baseline and at multiple time points after vaccination to measure vibriocidal antibody titers. For Arm A, vibriocidal GMT significantly increased nine days after the first dose with a response rate of 61–70%; however, early vibriocidal response (day 4–6) was rare (22%) and were similar in those who had received vaccine four years earlier and those who did not. Ogawa, but not Inaba, vibriocidal responses were higher on day 9 for those who had been vaccinated earlier, but vibriocidal titers during subsequent timepoints were similar in all study arms. In Arm B, there were no early response (day 4–6) after the second dose. Overall, OCV induced a measurable immune response by day 9, but no evidence of a rapid booster response was detected following revaccination. Since there was no evidence that the earlier vaccination stimulated

**Data availability statement:** Data presented or used for the analysis in the present manuscript are available as "S1 Data".

**Funding:** This work was supported by a grant from the Wellcome Trust to JA (grant number: 215663/Z/19/Z) and a grant from the National Institute of Allergy and Infectious Disease to DAS (5R01AI123422). The Wellcome Trust provided suggestions as to the study design, but neither funder had any role in data collection and analysis, decision to publish, or preparation of the manuscript.

**Competing interests:** The authors have declared that no competing interests exist.

a booster response when vaccination is repeated, this suggests that persons being revaccinated should receive the two doses as was recommended initially.

**Trial registration:** Pan African Clinical Trials Registry under the reference PACTR202102660004195.

---

## Background

Cholera continues as a major public health threat, especially in Sub-Saharan Africa [1]. Oral Cholera Vaccine (OCV) vaccination is one of the most effective interventions to prevent cholera [2–4]. While both killed whole cell and live attenuated vaccines have been licensed, only the killed oral whole cell vaccine is used through the global stockpile [5,6]. Since the duration of protection of OCV is estimated to be 3–5 years, the WHO recommends that revaccination may be needed after 3 years where there is continued risk of cholera [6]. To date, the vaccine manufacturers have not issued any recommendations for booster vaccination for the killed whole cell cholera vaccine.

Previous studies examined the response following a repeated (booster) dose with killed OCV (Shanchol). One study in Bangladesh found that children age < 5 years had a higher serological response to a single dose of OCV as measured by geometric mean vibriocidal titers if they had received the vaccine three years earlier, but other age groups (age > 5 years and adults) did not show this enhanced response [7]. Another study from India found no difference in antibody response when comparing responses of those who had received OCV five years earlier to those who had not [8]. A study using Dukoral, another killed OCV that includes the B subunit in the vaccine, found that an earlier dose of vaccine stimulated an early antibody response to the B subunit 4–5 days after a booster dose that did not occur in participants being immunized for the first time [9].

Whether immunization with killed OCV will lead to a booster response when the vaccine is given again is not clear but it would seem to be important when planning to reimmunize populations that are being immunized again after a prolonged period. If the initial immunization enhances a subsequent immunization, perhaps a single dose for re-immunization may be sufficient; however, if the initial immunization does not lead to an enhanced response, a two-dose regimen would seem preferable. The earlier studies in Bangladesh and India were carried out in cholera endemic areas where natural exposure between vaccinations might have influenced results and a study in an area that was not endemic was needed.

Because of an outbreak in the Mogode Health District in the Far North region of Cameroon area in 2017, the Cameroon Ministry of Public Health and its partners carried out an OCV vaccination campaign. This campaign aimed to give two doses Shanchol to people aged ≥1 year old. A coverage survey conducted between the two rounds and after the second round estimated that 80.1% of the population had received two doses and 4.3% had received one dose [10]. Since the campaign, there have been no reports of cholera between 2017 and 2021, nor has there been any

administration of cholera vaccinations by the Ministry of Health that may have confounded this analysis. This previous exposure of the populations of Mogode Health district to OCV vaccination provided an opportunity to explore the immune response of a booster OCV vaccination to answer the following questions: 1) When given OCV, do people who had received two OCV doses four years earlier develop a rapid (within 4–6 days) serum vibriocidal response more commonly than those who did not receive OCV four years earlier. 2) When given OCV, is the geometric mean titer (GMT) vibriocidal in each age group (1–4, 5–14, ≥ 15) superior among people who received OCV four years earlier compared to those who did not receive OCV four years earlier? 3) When given OCV, is the serum vibriocidal responses in children inferior to the response in older age groups? 4) Do the immune responses after one or two doses OCV administration differ in people exposed to one or two doses of OCV earlier?

Findings from this study are expected to guide the use of OCV when given to groups of people who were previously vaccinated.

## Materials and methods

### Ethics statement

Permission from adequately informed community authorities was obtained before accessing eligible participants in households. All study participants were fully informed in an appropriate language about study justifications, objectives, procedures, implications, and expected results and were given the time to ask questions before deciding to consent or not to participate. Only those that provided their consent were included. For minor participants (<21 years), parental permission was obtained. For minors aged between 12 and 20, their assent was sought and obtained prior to their inclusion in the study. Their assent was accompanied by a parental consent. Measures were taken to ensure confidentiality of patient data during inclusion, administration of interventions, patient sampling, data processing and analysis. All participants were observed 30 minutes after vaccine administration and visited to detect, report and investigate any adverse events occurring after vaccine administration. Similarly, measures were taken regarding the risk of infection, trauma and pain resulting from biological sampling. The research protocol was approved by the Cameroon National Ethics committee for Human Health Research (N°: 2020/12/1321/CE/CNERCH/SP).

### Trial registration

The study protocol was registered in Pan African Clinical Trials Registry under the reference PACTR202102660004195 (https://pactr.samrc.ac.za/TrialDisplay.aspx?TrialID=14625).

### Setting

The study was conducted in Mogode health district located in the Far North region of Cameroon where a two-round OCV campaign was conducted in May and June 2017. The immunization coverage estimated from the end campaign survey was estimated for two, one and zero doses documented coverage at 80.1%, 4.3% and 3.8% respectively [10]. Enrollment of participants for the present trial was conducted from February to June 2021. Between the 2017 campaign and the start of the study in 2021, no cholera cases were detected in Mogode Health District or any of bordering health districts by the national Ministry of Health surveillance system. Also, there was no OCV given in the areas since the 2017 campaign.

### Study design

Individuals living in Mogode Health District aged four years and above and who had received zero, one or two doses of OCV during the 2017 OCV campaign were eligible to participate. Those fulfilling inclusion criteria were randomized to Arm A or Arm B. Those in Arm A received a second dose on day 28, but Arm A received a second dose on day 14. Both

Arms were stratified into age groups (4 years, 5–14 years, ≥ 15 years) and OCV exposure status in 2017 (zero dose, one dose, two doses). Serum samples for Arm A were collected at baseline and on day 4–6, 9–12, 14–16 and 28–30 following administration of the first dose. Serum samples were collected for Arm B at baseline and on day 4–6, 9–12, 14–16 and 28–30 after the second dose. The intent was to determine and compare the serum vibriocidal responses overall, and by age group depending on the receipt of vaccine in 2017 following the first and second dose of vaccine when given four years later.

## Participants

All individuals aged 1 year and above during the 2017 OCV vaccination campaign living in the Mogode health district and who received either two, one or zero doses of OCV during the 2017 campaign were eligible. We excluded individuals who reported having received an OCV vaccine after the 2017 OCV campaign, women who were pregnant, persons enrolled in another study, those with a history of diarrhea during the seven days prior to first dose of vaccine (defined as ≥3 loose stools in 24 hours) or history of chronic diarrhea (lasting for more than 2 weeks in the past 6 months), those currently using laxatives, antacids, or other agents to lower stomach acidity, those with chromic health conditions, or immunosuppression and persons not willing to provide informed consent.

## Study procedures

**Community involvement.** With the help of local health authorities, the traditional community leaders were invited to a meeting at which the rationale, objectives, implementation procedures and expectations of the project were presented prior to intervention implementation and discussed. The leaders understood the project, and each made a commitment to facilitate the project's activities in their respective communities. They contributed to the identification of community volunteers to be involved in project activities in their communities. These included persons able to speak local (Kapsiki) and an official language (English or French) living in one of targeted communities. These community volunteers were trained to visit households, introduce the project to the head of household, and seek his permission to invite members of their household to visit the project site as potential participants. The community volunteers were also involved in reminding the people included in the study of their visit schedules.

**Enrollment.** Households in the Mogode health area in the Mogode health district were visited by community volunteers to introduce the project to household heads and obtain permission to brief household inhabitants and invite those eligible to the study site prepared for this purpose with the collaboration of the Mogode district hospital. Participants presenting themselves at the study site were informed about the study. Those who consented were screened for eligibility and selection criteria. Eligible participants meeting the inclusion criteria were assigned to the two arms of the study by randomization as described below. During the follow-up period, participants were reminded by telephone or through home visits of the dates and times of their follow-up visits. Enrollment of participants for the present trial was conducted from February to June 2021.

**Determining participants age, anthropometric values and prior vaccination status.** For eligible participants, the age was collected on the identity card or birth certificate. In the absence of these documents, age was tracked using the documented ages of other children in the household or neighborhood, or using major events whose timing in the past was known. Vaccination status 4 years prior to OCV was determined using the vaccination record, or tracked using photos of the vaccine, the vaccine administration protocol and specific questions asked to participants or guardians (the period of the campaign, the presence of the participant in the community during the campaign, the description of the vaccine vial and the taste of the vaccine if possible). Weight and height were measured for all participants at enrollment using standardized procedures and calibrated instruments. Weight was measured using a digital scale for all participants (aged minimum 4 years). Height was measured to the nearest 0.1 cm using a portable stadiometer. Nutritional status classification was not performed.

**Global Public Health**

**Randomization**. Participants enrolled in the study were randomized into two arms, A or B and each arm was stratified by age group and prior exposure status to OCV. The team responsible for randomization was different from the team enrolling and randomly assigning participants to study arms.

## Interventions

The administration of the vaccine was conducted at the study implementation site. Participants in arm A received oral administration of one dose of OCV (Shanchol, Lots 1: SCN006A19 and Lot 2: SCN034A20) on day zero. Participants in arm B received two doses of OCV (same batch); the first dose was on day 0 and the second dose was on day 14–16. Shanchol consists of a mixture of killed *V. cholerae* O1 and O139 in single-dose glass vial containing 1.5 mL of vaccine. The entire vial of OCV was administered orally to each participant irrespective of their age. The vaccine vial was shaken to ensure mixing and was inspected by the study nurse before administration. To monitor for safety, all participants were observed for 30 minutes after taking the vaccine and had to report any adverse events occurring within two weeks of receiving the vaccine to the community worker or during the next study planned visit. For ethical reasons, and to ensure that all study participants had an opportunity to receive the recommended two doses of vaccine, the participants in arm A were invited to take the second OCV dose after their follow-up period, which ended on day 28, but no additional serum samples were collected after this second dose in arm A.

## Outcome assessment

In arm A, participants were followed up for 28 days with five blood draws and participants in arm B, were followed up for 42 days with six blood draws. The timing of the follow-up vaccinations and blood draws along with the acceptable visit windows for each event are shown in Fig 1. The primary outcome was vibriocidal geometric mean titer (GMT) and sero-conversion rate measured four to six days following the first dose of OCV in Arm A and the four to six days after the second dose in Arm B to assess and compare early immune responses between participants exposed to two doses OCV four years earlier to those exposed to zero doses four years earlier. Secondary outcomes included 1) Age group specific (4, 5–14, ≥ 15 years) and overall vibriocidal GMT and seroconversion rates measured at days 4–6, 9–12, 14–16 and 28–30 following OCV administration to compare vibriocidal responses between participants who received one or two OCV doses four years earlier and those who received zero doses four years earlier. 2) Vibriocidal GMT and seroconversion rates (≥ 4-fold rise in vibriocidal titers compared with baseline) measured four to six days following a single OCV dose to compare early immune response between participants exposed to one or two doses of OCV four years earlier and those exposed to zero doses. 3) Vibriocidal GMT and seroconversion rate 14–16 days following exposure to the single dose (arms A and B) and days 18–20, 23–25, 28–30 and 42–44 for arm B.

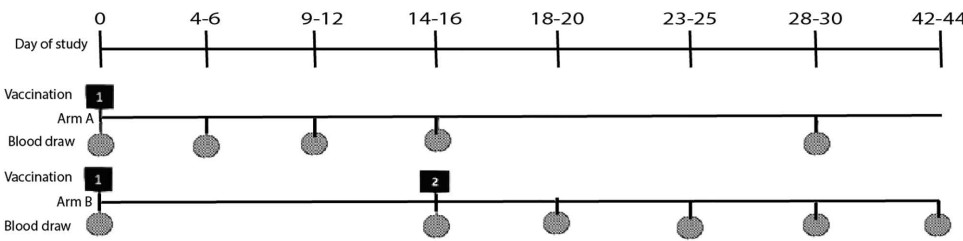

**Fig 1. Visit windows, vaccination and blood draw schedules.**

## Laboratory procedures

The timing of the blood samples is shown in Fig 1. Blood sample collection procedures depended on age group. For age group <5years, a fingerstick sample was obtained using a lancet and capillary tube to collect 350 µl of blood which was then diluted in 1400 µl normal saline which resulted in a 1:5 dilution of whole blood. For participants ≥5 years 5 ml of venous blood was collected using a syringe. The blood samples were centrifuged, and the serum was stored at -20°C until assayed.

For the vibriocidal assay, we used *V. cholerae* strains isolated from Cameroon (Inaba 600040 and Ogawa CPC-BACT-01) as previously described [11]. The strains were incubated with serially diluted, heat-inactivated serum with exogenous guinea pig complement (Sigma Aldrich, catalogue # S234395) on a shaker (50 rev/min) at 37°C for 1 hour [7].The starting dilution for the previously undiluted serum was 1:10 while the previously diluted serum from young children was used without further dilution considering that the 1:5 dilution of whole blood to be equivalent to the 1:10 dilution of serum. Vibriocidal titers were defined as the reciprocal of the highest serum dilution which results in a 50% reduction in optical density (595 nm) compared with growth control without serum. Standard sera, comprising of high titer human serum and high titer rabbit serum were employed to normalize the outcomes and mitigate the effects of inter-experimental variation. Samples were tested in duplicates. Titers lower than the lowest detectable titer were assigned to have a titer equivalent to half of the lowest detectable titer. The threshold for inter- and intra-experimental variation was set at 2-fold. Seroconversion was defined as ≥ 4-fold rise in vibriocidal titers compared with baseline titer.

## Sample size

To determine whether a rapid vibriocidal response following single dose OCV administration among those exposed to OCV four years earlier is superior to those exposed to zero doses four years earlier, we attempted to recruit 20 people for each age group (4, 5–14, ≥ 15 years old) in Arm A and the same number for Arm B. Among those who had received OCV four years earlier, we intended to evaluate each age group into those who had received zero, one dose or two doses earlier. Thus, total sample size was estimated to be 360 with 180 participants expected per study arm, including 60 per OCV vaccination status (0, 1 or 2 doses) distributed in 20 per age group. We hypothesized that an early vibriocidal response following single dose administration (Arm A) will occur among 25% of those in non-exposed group (zero OCV dose four years earlier) and will be increased to 75% in the exposed group (exposed to 2 OCV doses four years earlier) (confidence = 95%, power = 90%), that early immune response in arm A will not differ to that of Arm B and that the dropout rate in each study arm will be 10% or less.

## Data analysis

To identify confounders between OCV exposure status and between study arms, we compared the distribution of participants' baseline characteristics between those exposed to zero OCV doses and those exposed to two doses four years earlier, and between those assigned to the two arms. For quantitative variables with normal distribution, mean was used as measure of central tendency while with variables with anormal distribution, median and IQR were used to summarize the variables.

For the primary analysis, we compared the vibriocidal GMT and seroconversion rates on day 4–6 following the administration of a single dose of OCV between participants exposed to two OCV doses in 2017 and those exposed to zero doses using Wilcoxon and chi-square tests respectively. For the estimate of vibriocidal seroconversion rate, logistic regression model was used for the adjustment using individual and household characteristics as adjustment variables.

For the secondary analyses, we carried out the same analysis per age group between participants exposed to 1 or 2 OCV doses and those exposed to zero doses in 2017. These estimates were also conducted on day 9–12, 14–16, and 28–30 following the single OCV dose administration for participants of arm A. Diagrams showing estimates of vibriocidal GMTs with a 95% confidence interval on the planned blood draw days following a single dose OCV administration (Arm A)

and two doses OCV administration (arm B) was generated for overall, by age group and according to 2017 OCV vaccination status. The comparison of vibriocidal GMT was done using the Wilcoxon test and that of seroconversion rate using the chi square test.

To compare the statistical significance of the vibriocidal seroconversion rate between groups (Arm A and Arm B as well as for other desegregation), we employed a multivariable logistic regression model. The model was used to test the null hypothesis of no difference in seroconversion status (dependent variable) based on vaccination status in 2017 (main independent variable).

To ensure the validity of the p-values, the model adjusted for potential confounding factors at both the individual and household levels, including sex, age group (categorized for all ages), floor material, and access to improved toilet facilities, as well as weight and height.

The comparisons were made with one tailed 95% confidence interval. Analyses were conducted using Stata (Version 18).

## Results

### Enrollment

The consort flow diagram for the study is shown in Fig 2. Of the 430 people who were received in the study site, 3 did not consent to be screened and 77 did not meet inclusion criteria. 350 (81.4%) participants were enrolled in the study with 136, 101 and 113 with OCV vaccination status four years earlier of 0, 1 and 2 doses respectively. For those who were included, 174 and 176 were randomly assigned to Arm A and Arm B respectively.

**Baseline characteristics of participants regarding study arms.** Table 1 presents the comparison of participants' characteristics per study arm. No difference in the distribution of these characteristics was detected.

**Comparison of characteristics of participants per OCV exposure status, four years earlier.** Table 2 shows a comparison of the distribution of the characteristics of participants exposed to two or at least one dose of OCV compared with those exposed to zero doses. The comparison reveals some differences in distribution for age groups, and housing materials between those who did and did not receive OCV earlier.

**Comparison of the early (day 4–6) vibriocidal response in participants who received zero, two doses or at least one dose (one and two doses) of OCV four years earlier.** The early immune responses (vibriocidal GMT and seroconversion rate) of participants (all age group, per age group and for serotypes Inaba and Ogawa) exposed to two or at least one dose of OCV versus those exposed to zero dose is presented in Table 3. An early (day 4–6) rise in vibriocidal titers was rare among all participants including those who received OCV earlier and those who did not, and there was no difference between the groups. This was consistent across age groups and serotypes (Ogawa and Inaba).

**Distribution and comparison of vibriocidal GMT and seroconversion rates 9–12 days after a single OCV dose depending on receipt of OCV four years earlier.** Table 4 presents the comparison of vibriocidal GMT and seroconversion rate 9–12 days following single OCV dose depending on receipt of OCV earlier. The Ogawa vibriocidal GMT at day 9–12 following a single OCV dose was significantly higher among all ages, and for each age group for the participants who received OCV four years earlier compared to those who received zero doses. A similar tendency was observed for Inaba serotype but the difference was not significant. An increase in vibriocidal seroconversion rates for Ogawa and Inaba was also observed but the differences were not significant. Between those who received OCV earlier and those who had not.

**Trends of vibriocidal GMT following single dose administration of OCV.** As shown in Figs 3 and 4, the vibriocidal GMT for Inaba serotype for the entire group and for each of the age groups for days 9–12, 14–14 and 28–30 days following single OCV dose administration was not higher in participants who received two OCV doses earlier, or who received either one or two OVC doses earlier compared to those who did not receive OCV earlier. However, the vibriocidal GMT for Ogawa serotype for these days following single OCV dose administration was higher in participants exposed to

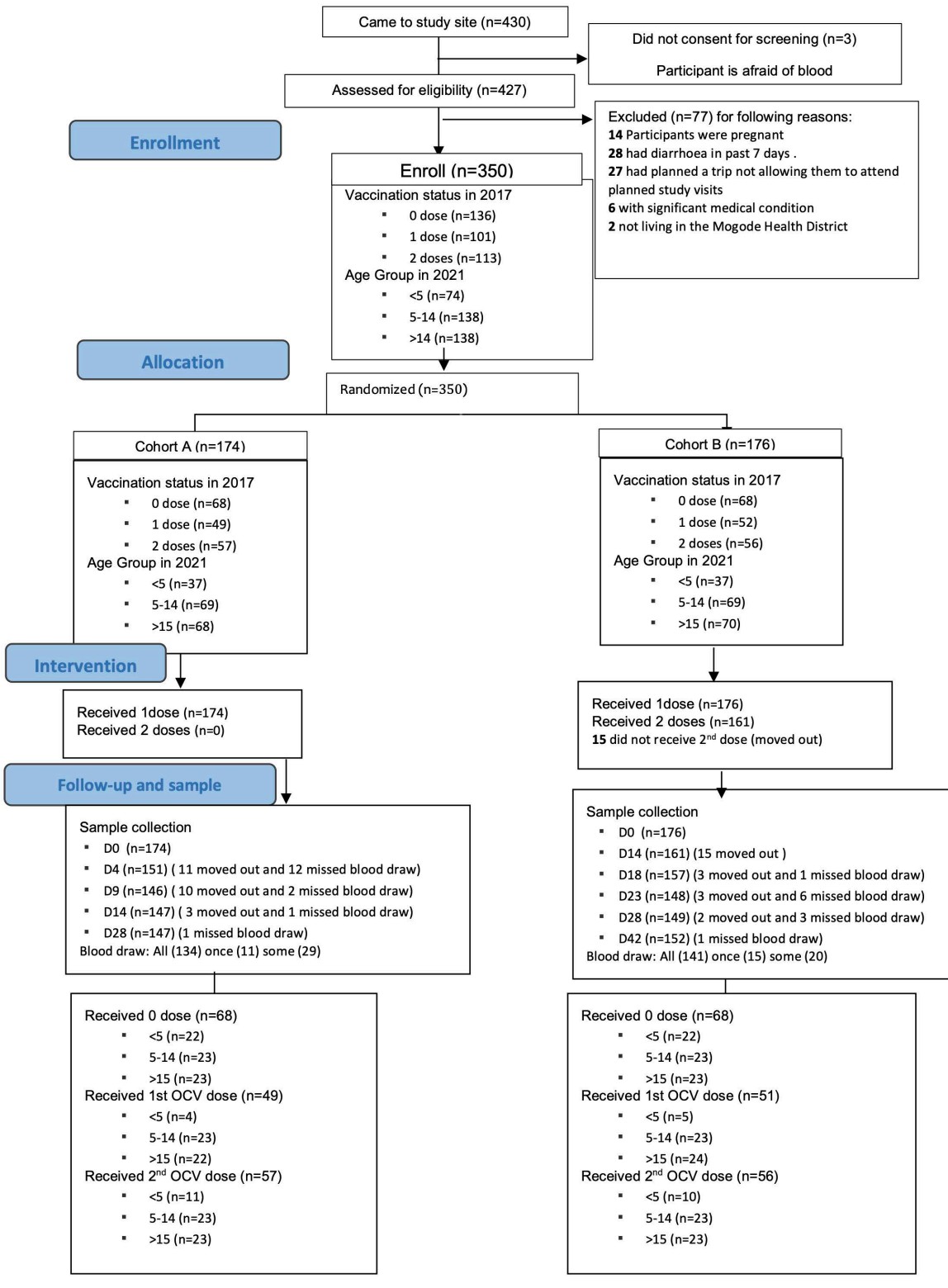

**Fig 2. Consort flow diagram.**

PLOS Global Public Health

**Table 1. Distribution of participants characteristics per study arm.**

| Characteristics | Arm A (N = 174) | Arm B N(176) |
| --- | --- | --- |
| Sex, n(%) | | |
| Male | 72 (41.4) | 69 (39.2) |
| Age group in years,n(%) | | |
| 4 | 37 (21.3) | 37 (21.0) |
| 5-14 | 69 (39.7) | 69 (39.2) |
| ≥15 | 68 (39.1) | 70 (39.8) |
| Age median [QR1, QR3] | | |
| | 11 [6, 29] | 12 [6, 31.5] |
| Floor material of participants' houses, n (%) | | |
| Cement/ Brick/Tiles | 50 (28.7) | 40 (22.7) |
| Using improved toilet facility, n (%) | | |
| | 19 (10.9) | 15 (8.5) |
| Weight in kg, median [QR1, QR3] | | |
| | 28 [16, 50] | 30.5 [18, 52] |
| Height in meters, median [QR1, QR3] | | |
| | 1.4 [1.1, 1.6] | 1.4 [1.1, 1.6] |
| Body Mass Index, median [QR1, QR3]* | | |
| | 17.3 [15.1, 20.7] | 18.3 [14.9, 20.7] |
| Baseline LVGMT Inaba, geometric mean [CI] | | |
| | 1[0.9,1.1] | 1 [1,1.1] |
| Baseline LVGMT Ogawa, geometric mean [CI] | | |
| | 1.5[1.4,1.6] | 1.4[1.3,1.5] |
| Vaccination status, n (%) | | |
| 0 dose | 68 (39.1) | 68 (38.6) |
| 1 dose | 49 (28.7) | 52 (29.6) |
| 2 doses | 57 (32.8) | 56 (31.8) |

LVGMT = log of vibriocidal Geometric mean titer, VSCR = vibriocidal seroconversion rate. *
Analysis includes only units aged 5 years and above.

two OCV doses, or one or two OCV doses four years earlier than those who did not receive OCV earlier. This trend was observed for the entire group and for each age group.

**Trends of vibriocidal GMT and seroconversion in study arms A and B.** The GMT for the two study arms is shown in Fig 5 and 6. For both Ogawa and Inaba serotypes, for all ages and for each age group, the administration of the second dose of OCV (arm B) did not lead to an increase in GMT vibriocidal values. The vibriocidal GMT titres of arm A was not different from arm B in participants and there was no difference depending on receipt of OCV four years earlier.

**Comparison of Vibriocidal GMT after a single dose of OCV after receipt of two OCV four years earlier between children and adults.** Table 5 presents the comparison of vibriocidal GMT and seroconversion rate 4 days and 9–12 days following single OCV dose for children and older participants that received two doses OCV earlier. For the Ogawa and Inaba serotypes, the tendency suggests a higher response in children (<5 years) but with no statistical significance. The GMT between children and older participants is presented in Fig 7.

**Comparison of Vibriocidal GMT and Seroconversion rates 28–30 days between arm A and arm B.** Table 6 presents the comparison between arm A and arm B of Vibriocidal GMT and Seroconversion rates 28–30 days compared to day 0. There was no difference between arms A and B for the entire groups or for each age group or serotype.

**Table 2. Baseline comparison of participants who received no OCV earlier and those who received one or two doses earlier.**

| Characteristics | Received zero doses earlier compared to those who received two doses earlier | | | Zero doses compared to those who received one or two doses earlier | |
|---|---|---|---|---|---|
| | Zero dose (N=136) | Two doses(N=113) | p.value | One or two doses (N=214) | P value |
| Sex, n(%)* | | | | | |
| Male | 53 (39) | 48 (42.5) | 0.57 | 88 (41.1) | 0.68 |
| Age group n (%)* | | | | | |
| 4 | 44(32.4) | 21 (18.6) | 0.048 | 30 (14.) | 0.00 |
| 5-14 | 46 (33.8) | 46 (40.7) | | 92 (43) | |
| ≥15 | 46 (33.8) | 46 (40.7) | | 92 (43) | |
| Age, median[QR1, QR3] | | | | | |
| | 9 [5, 25] | 12 [6, 29] | 0.045 | 13 [7, 36] | 0.00 |
| Floor material, n (%)* | | | | | |
| Cement/ Brick/Tiles | 47 (34.6) | 17 (15) | 0.00 | 43 (20.1) | 0.003 |
| Using improved toilet facility,n(%)* | | | | | |
| Yes | 18 (13.2) | 9 (8) | 0.18 | 16 (7.5) | 0.08 |
| Weight in kg, median [QR1, QR3]++ | | | | | |
| | 24 [15, 50] | 30 [17, 55] | 0.05 | 35 [19, 53] | 0.004 |
| Height in m, median [QR1, QR3]++ | | | | | |
| | 1.3 [1, 1.6] | 1.4 [1.1, 1.6] | 0.05 | 1.5 [1.2, 1.6] | 0.002 |
| Body Mass Index, mean [CI]++ | | | | | |
| | 17.4 [15.1, 20.6] | 17.4 [15, 21.3] | 0.7 | 17.5 [15, 20.7] | 0.9 |
| Baseline VGMT inaba, [CI]++ | | | | | |
| LVGMT Inaba | 1[0.9,1.1] | 1 [0.9,1.1] | 0.53 | 1 [1,1.1] | 0.21 |
| VGMT Inaba | 9.3[7.6,11.5] | 9.8 [7.9,12.2] | | 11.2 [9.3,13.4] | |
| Baseline VGMT Ogawa, [CI]++ | | | | | |
| LVGMT Ogawa | 1.3[1.2, 1.5] | 1.5 [1.4, 1.7] | 0.02 | 1.5[1.4, 1.6] | 0.007 |
| VGMT Ogawa | 21.2[15.8, 28.3] | 34.5 [24.6, 48.5] | | 34.5[27.0, 44.0] | |

++P values are from Wilcoxon test, * P values are from chi-square, VGMT=vibriocidal Geometric mean titer.

**Detected adverse events following OCV administration.** Five adverse events were detected including 2 cases of nausea in the 30 minutes following immunization and 03 cases of diarrhea in the fourteen days following OCV administration. None of these cases was serious.

## Discussion

The present study was intended to identify a rapid response to a dose of OCV if the subject had received the OCV vaccine earlier indicating that these people had responded with a booster response. However, we did not observe rapid responses in those who had received OCV earlier. In fact, an early response was seen rarely in any of the groups and there was not a significant difference in the proportion with a rapid response between those who had received OCV earlier and those who did not. This finding was consistent for each of age groups (4, 5–14, ≥ 15 years old) and for both Ogawa and Inaba vibriocidal assays.

We did observe a higher peak Ogawa vibriocidal GMT after the initial dose of OCV but not a higher Inaba GMT in those who had received OCV earlier compared to those who had not received OCV earlier. This was observed both for the entire age group and for age specific groups.

**Table 3. Comparison of Vibriocidal GMT and Seroconversion rates 4-6 days after a single dose of OCV depending on receipt of OCV four years earlier.**

| Age group and outcome (VGMT and VSCR) | Early vibriocidal titer changes among those who received zero doses of OCV and those who received two doses earlier (Arm A) | | | Early vibriocidal titer changes among those who received zero doses of OCV and those who received one or two doses earlier (Arm A) | |
|---|---|---|---|---|---|
| | Zero dose | Two doses | P value (adjusted) | At least one dose (One and two doses) | P value (adjusted) |
| **All age group** | **N=59** | **N=49** | | **N=92** | |
| LVGMT Inaba,[CI]+ | 1.2 [1;1.4] | 1.2 [1;1.4] | 0.7 | 1.1 [1;1.3] | 0.8 |
| LVGMT Ogawa,[CI]+ | 1.5 [1.3;1.7] | 1.7 [1.5;1.9] | 0.1 | 1.6 [1.5;1.8] | 0.2 |
| VSCR Inaba, %[CI]* | 23.7 [12.5;34.9] | 22.4 [10.3;34.6] | 0.9 (0.60) | 12 [5.2;18.7] | 0.1 (0.1) |
| VSCR Ogawa, %[CI]* | 15.3 [5.8;24.7] | 18.4 [7.1;29.6] | 0.70 (0.5) | 10.9 [4.4;17.4] | 0.4 (.60) |
| **4 years** | **N=17** | **N=9** | | **N=13** | |
| LVGMT Inaba,[CI]+ | 1.2 [0.9;1.6] | 1.3 [0.7;1.9] | 0.6 | 1.1 [0.7;1.5] | 1 |
| LVGMT Ogawa,[CI]+ | 1.2 [0.8;1.5] | 1.5 [1;2] | 0.2 | 1.4 [.9;1.8] | 0.4 |
| VSCR Inaba, % [CI]* | 29.4 [5.3;53.6] | 33.3 [0;71.8] | 0.8(0.9) | 23.1 [5.3;49.6] | 0.70 (0.70) |
| VSCR Ogawa, % [CI]* | 29.4 [5.3;53.6] | 44.4 [3.9;85] | 0.4(.3) | 30.8 [5.3;59.8] | 0.9 (0.60) |
| **5-14 years** | **N=22** | **N=22** | | **N=42** | |
| LVGMT Inaba,[CI]+ | 1.2 [.8;1.5] | 1 [0.8;1.2] | 0.3 | 1 [0.8;1.2] | 0.3 |
| LVGMT Ogawa,[CI]+ | 1.4 [1.2;1.6] | 1.5 [1.2;1.8] | 0.9 | 1.5 [1.3;1.7] | 0.6 |
| VSCR Inaba, % [CI]* | 22.7 [3.7;41.7] | 18.2 [0.7;35.7] | 0.70(0.3) | 9.5 [3.7;18.8] | 0.1 (0) |
| VSCR Ogawa, % [CI]* | 18.2 [0.7;35.7] | 13.6 [0;29.2] | 0.70(.2) | 7.1 [.70;15.3] | 0.2 (0) |
| **> 14 years** | **N=20** | **N=18** | | **N=37** | |
| LVGMT Inaba,[CI]+ | 1.2 [.8;1.5] | 1.4 [1.1;1.8] | 0.2 | 1.3 [1.1;1.6] | 0.4 |
| LVGMT Ogawa,[CI]+ | 1.8 [1.4;2.2] | 2.1 [1.7;2.5] | 0.3 | 1.8 [1.5;2.1] | 0.9 |
| VSCR Inaba, % [CI]* | 20 [.8;39.2] | 22.2 [0.9;43.5] | 0.9(0.2) | 10.8 [0.8;21.3] | 0.3 (1) |
| VSCR Ogawa, % [CI]* | 0 [0;0] | 11.1 [0;27.2] | 0.1 | 8.1 [0;17.3] | 0.2 |

+ P.value are from Wilcoxon test, * P.value are from chi-square, LVGMT = log of vibriocidal Geometric mean titer, VSCR = vibriocidal seroconversion rate. Adjusted = p.value from logistic regression. The final p.value column is from the comparison between the zero-dose group and the "at least one dose" group (fourth column with data).

In Arm B, there were no rapid responses on day 18 (4–6 days after dose 2) to suggest a booster response to this second dose. The GMT at day 14, 28-days following administration of one dose of OCV in arm A and two doses 14 days apart in arm B did not differ for overall participants, for each age group, serotype Ogawa and Inaba and by prior exposure to OCV vaccine status.

The administration of two doses Oral Cholera Vaccine (OCV) is known to protect about 65% and a lower percentage of children aged 1–4 years for five years following the vaccination [6]. The present study was planned with the hypothesis that the magnitude and speed of the immune response following OCV vaccination would correlate with immune memory and would therefore be higher and faster in previously vaccinated individuals than in those not previously vaccinated. The vibriocidal GMT and seroconversion rate for the Ogawa and Inaba serotypes was not higher for overall participants nor for each of the three age groups among those exposed four years earlier to one or two doses compared to those exposed to zero OCV dose four years earlier. These results are consistent with those of a study conducted in Bangladesh measuring immune response in all age groups on day 3 following administration of a single dose of Shanchol vaccine [7]. The present study is the first study to examine these responses in these age groups, ≤ 5 years, 5–14 and ≥ 15 years) in people who had received two doses of OCV vaccine 4 years earlier. It is unlikely that not detecting differences of GMT and response rates between participants exposed to one or two doses of OCV and those exposed to zero doses be explained

**Table 4. Comparison of Vibriocidal GMT and Seroconversion rates 9-11 days after a single dose of OCV depending on receipt of OCV four years earlier.**

| Blood characteristics (outcomes) 9–11 days after the single dose administration | Arm A Number of OCV doses exposed four year earlier (zero dose and two doses) | | | Arm A Number of OCV doses exposed four year earlier (zero dose and one or two doses) | |
|---|---|---|---|---|---|
| | Zero dose | Two doses | P-value (adjusted) | Atleast one dose (One or two doses) | P-value (adjusted) |
| All age group | N = 59 | N = 47 | | N = 87 | |
| LVGMT Inaba,[CI]+ | 1.8 [1.6;2] | 1.9 [1.7;2.2] | 0.5 | 1.9 [1.7;2.1] | 0.7 |
| LVGMT Ogawa,[CI]+ | 1.9 [1.8;2.1] | 2.6 [2.4;2.8] | 0 | 2.4 [2.2;2.5] | 0 |
| VSCR Inaba, %[CI]* | 62.7 [50;75.4] | 68.1 [54.3;81.9] | 0.60 (0.4) | 64.4 [54.1;74.6] | 0.8 (0.60) |
| VSCR Ogawa, %[CI]* | 61 [48.2;73.8] | 70.2 [56.6;83.8] | 0.3 (0.2) | 65.5 [55.3;75.7] | 0.60 (0.3) |
| 4 years | N = 19 | N = 8 | | N = 11 | |
| LVGMT Inaba,[CI]+ | 2 [1.6;2.4] | 2.5 [1.8;3.2] | 0.2 | 2.2 [1.6;2.8] | 0.6 |
| LVGMT Ogawa,[CI]+ | 1.8 [1.4;2.2] | 2.6 [2;3.1] | 0 | 2.4 [1.9;2.9] | 0.1 |
| VSCR Inaba, % [CI]* | 73.7 [51.9;95.5] | 100 [100;100] | 0.1 | 90.9 [51.9;111.2] | 0.3 (1) |
| VSCR Ogawa, % [CI]* | 73.7 [51.9;95.5] | 100 [100;100] | 0.1 | 90.9 [51.9;111.2] | 0.3 (.60) |
| 5-14 years | N = 20 | N = 21 | | N = 42 | |
| LVGMT Inaba,[CI]+ | 1.8 [1.5;2.1] | 1.9 [1.5;2.3] | 0.8 | 1.9 [1.7;2.2] | 0.6 |
| LVGMT Ogawa,[CI]+ | 2.1 [1.9;2.3] | 2.5 [2.2;2.9] | 0 | 2.4 [2.2;2.6] | 0 |
| VSCR Inaba, % [CI]* | 60 [36.5;83.5] | 71.4 [50.4;92.5] | 0.4(1) | 73.8 [36.5;87.7] | 0.3 (0.70) |
| VSCR Ogawa, % [CI]* | 80 [60.8;99.2] | 71.4 [50.4;92.5] | 0.5(0.5) | 66.7 [60.8;81.5] | 0.3 (0.3) |
| > 14 years | N = 20 | N = 18 | | N = 34 | |
| LVGMT Inaba,[CI]+ | 1.6 [1.3;2] | 1.8 [1.3;2.2] | 0.7 | 1.7 [1.4;2] | 0.7 |
| LVGMT Ogawa,[CI]+ | 1.9 [1.6;2.3] | 2.7 [2.3;3] | 0 | 2.4 [2.1;2.7] | 0.1 |
| VSCR Inaba, % [CI]* | 55 [31.1;78.9] | 50 [24.4;75.6] | 0.8(0.70) | 44.1 [31.1;61.7] | 0.4 (0.5) |
| VSCR Ogawa, % [CI]* | 30 [8;52] | 55.6 [30.1;81] | 0.1(0.3) | 55.9 [8;73.5] | 0.1 (0.1) |

+ P.value from Wilcoxon test, * P.value from chi-square, LVGMT = Log of vibriocidal Geometric mean titer, VSCR = vibriocidal seroconversion rate. Adjusted = p.value are from logistic regression. the final p.value column is from the comparison between the zero-dose group (first column with data) and the at least one dose group (fourth column with data).

by differences in the distribution of some characteristics between exposure status at baseline, since these differences were taken into account in the adjustment of the estimation of the comparison tests. Similarly, it is unlikely that repeated, low-grade or undetected natural exposure to Vibrio cholerae in the study setting may have reduced the differences in vibriocidal antibody responses between vaccinated and unvaccinated participants since our team conducted a year-long epidemiological surveillance of cholera, screening suspected cases using rapid diagnostic tests for cholera. During this surveillance period, no positive cases were detected.

A number of assumptions that may explain the lack of difference in early immune response according to the status of OCV vaccination status deserve to be reviewed. The relatively high frequency of malnutrition in the Far North Cameroun region, as documented by recent demographic health surveys, could explain the low immune response in people who have been previously vaccinated, but in this case, a difference in immune response in adults would have been observed [12]. The vaccination status of most of the participants was not based on vaccination cards as these cards were no longer available but determined using process guided by a previous study that was documented reliable in assessing immunization status in people vaccinated without cards [13]. Despite the fact that the association between vibriocidal antibody titers and cholera immunity is not absolute, the present result suggests that early immune response resulting from single OCV administration is not sufficient to ensure cholera protection in all age groups [14]. This trend

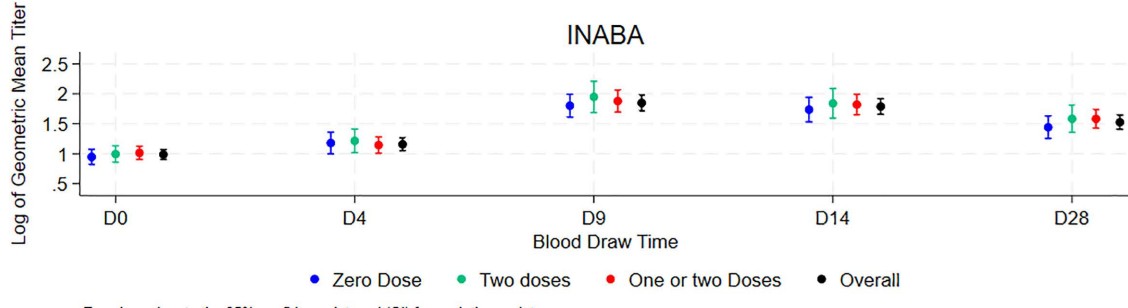

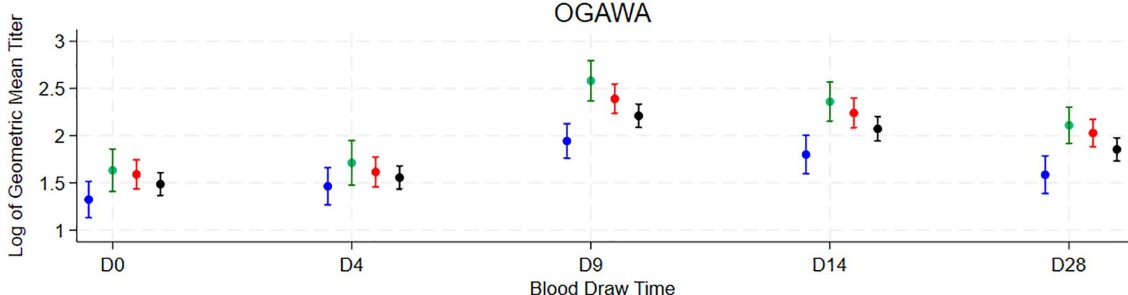

**Fig 3. Vibriocidal Geometric Mean Titer following single OCV of the three participating OCV status four year earlier through the span of the study.**

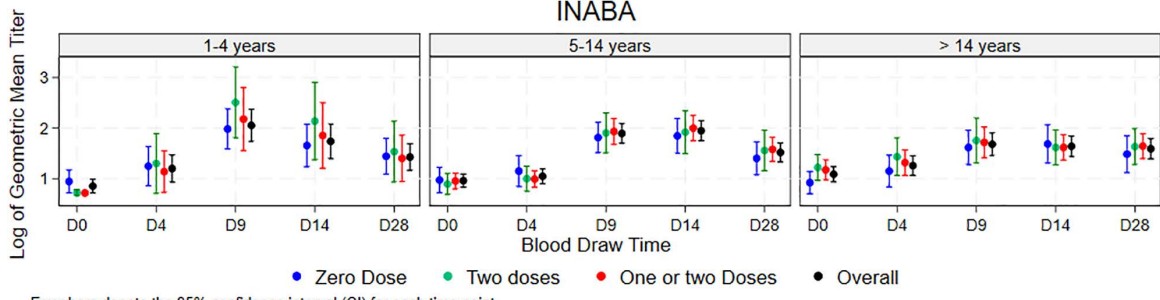

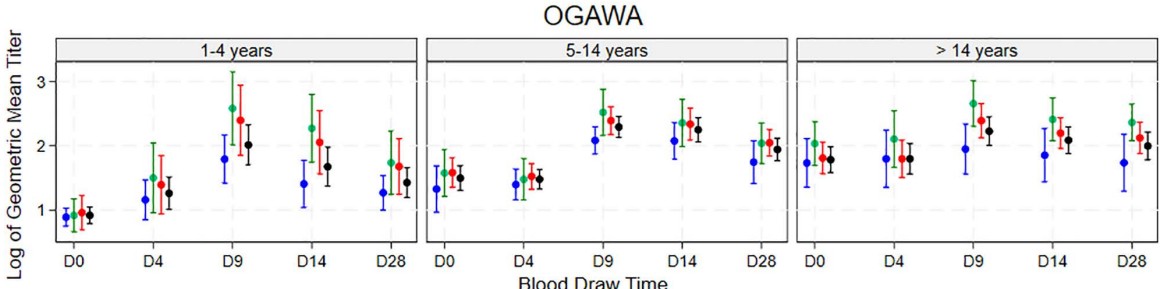

**Fig 4. Age stratified vibriocidal Geometric Mean Titer following single dose administration of the three participants OCV status four year earlier through the span of the study.**

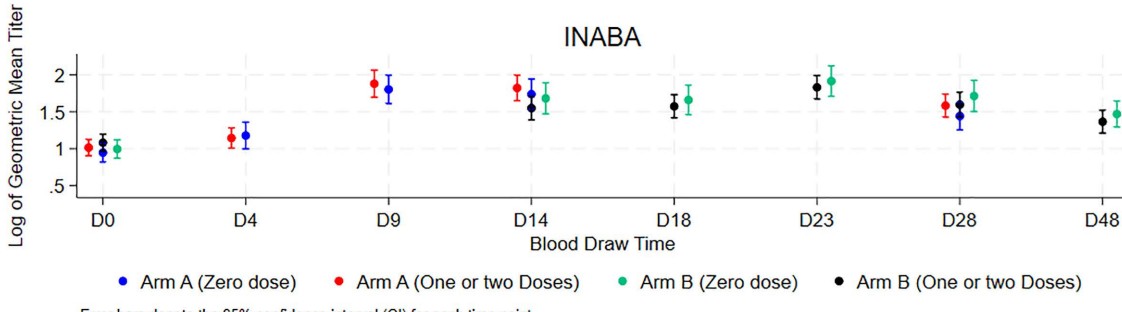

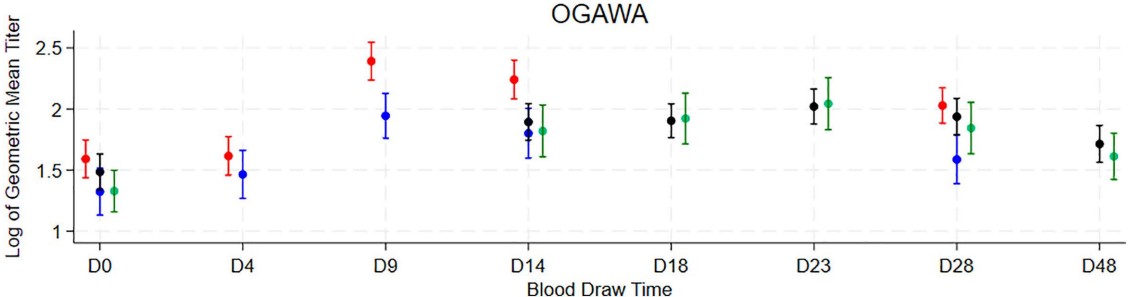

**Fig 5. Vibriocidal GMT of the participants who received 1 or 2 OCV doses earlier in Arms A and B through the span of the study.**

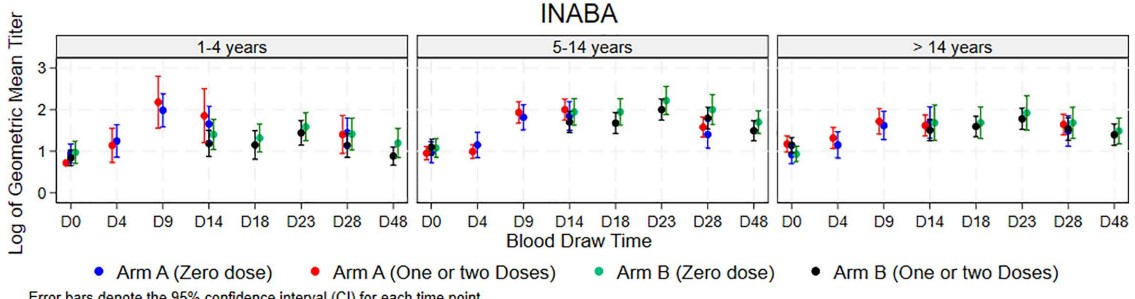

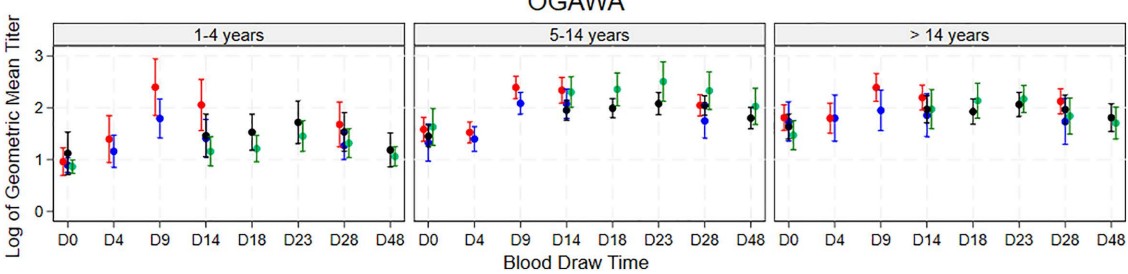

**Fig 6. Age stratified Vibriocidal GMT of the participants who received 1 or 2 OCV doses earlier in Arms A and B through the span of the study.**

**Table 5. Comparison between children and older participants of Vibriocidal GMT and Seroconversion rates 4 days and 9-11 days after a single dose of OCV for those that received two doses of OCV four years earlier.**

| Participants (2 doses four years earlier) | Immune response after first dose administration | | |
|---|---|---|---|
| | 4 years | At least 15 years | P.value (adjusted) |
| | Arm A immune response 4 days after the first dose administration | | |
| N | N = 9 | N = 18 | |
| LVGMT Inaba,[CI]+ | 1.3 [0.7;1.9] | 1.4 [1.1;1.8] | 0.7 |
| LVGMT Ogawa,[CI]+ | 1.5 [1;2] | 2.1 [1.7;2.5] | 0.1 |
| VSCR Inaba, %[CI]* | 33.3 [-5.1;71.8] | 22.2 [.9;43.5] | 0. (0.7) |
| VSCR Ogawa, %[CI]* | 44.4 [3.9;85] | 11.1 [-5;27.2] | 0. (0.2) |
| | Arm A immune response 9–11 days after the first dose administration | | |
| N | N = 8 | N = 18 | |
| LVGMT Inaba,[CI]+ | 2.5 [1.8;3.2] | 1.8 [1.3;2.2] | 0.1 |
| LVGMT Ogawa,[CI]+ | 2.6 [2;3.1] | 2.7 [2.3;3] | 0.7 |
| VSCR Inaba, %[CI]* | 100 [100;100] | 50 [24.4;75.6] | .(.) |
| VSCR Ogawa, %[CI]* | 100 [100;100] | 55.6 [30.1;81] | .(.) |

+ P-value from Wilcoxon test, * P-value from chi-square, LVGMT = Log of vibriocidal Geometric mean titer, VSCR = vibriocidal seroconversion rate. Adjusted = p.value are from logistic regression

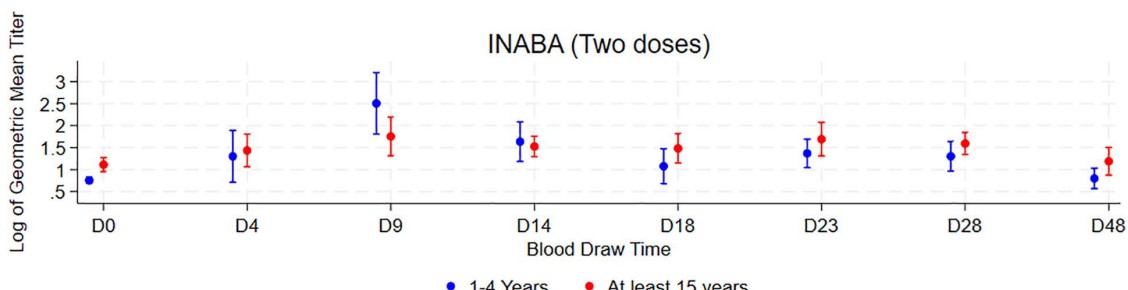

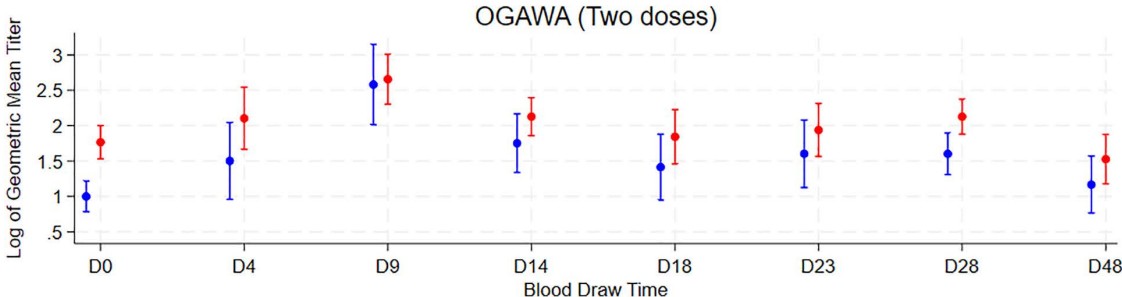

**Fig 7. Vibriocidal GMT of children (<5 years) and older participants (at least 15 years) who received 2 OCV doses earlier through the span of the study.**

should be confirmed exploring in the same context the use of other cholera vaccine immunity markers which appear in studies conducted in other contexts to show a higher booster immune response in populations previously exposed to vaccination [7].

**Table 6. Comparison of Vibriocidal GMT and Seroconversion rates 28-30 days after the first dose of OCV arm A and arm B.**

| Participants | Arm A and arm B immune response 28–30 days after the first dose administration | | |
| --- | --- | --- | --- |
| | Arm A | Arm B | P-value (adjusted) |
| All age group | N = 147 | N = 150 | |
| LVGMT Inaba,[CI]+ | 1.5 [1.4;1.6] | 1.6 [1.5;1.8] | 0.2 |
| LVGMT Ogawa,[CI]+ | 1.9 [1.7;2] | 1.9 [1.8;2] | 0.6 |
| VSCR Inaba, %[CI]* | 49.7 [41.5;57.8] | 50.3 [42.2;58.5] | 0.9 (0.8) |
| VSCR Ogawa, %[CI]* | 33.3 [25.6;41] | 43.3 [35.3;51.4] | 0.1 (0.1) |
| 4 years | N = 31 | N = 33 | |
| LVGMT Inaba,[CI]+ | 1.4 [1.2;1.7] | 1.3 [1.1;1.5] | 0.5 |
| LVGMT Ogawa,[CI]+ | 1.4 [1.2;1.7] | 1.4 [1.2;1.6] | 1 |
| VSCR Inaba, % [CI]* | 45.2 [26.6;63.7] | 31.3 [14.3;48.2] | 0.3 (0.2) |
| VSCR Ogawa, % [CI]* | 38.7 [20.5;56.9] | 45.5 [27.5;63.4] | 0.60 (0.60) |
| 5-14 years | N = 61 | N = 63 | |
| LVGMT Inaba,[CI]+ | 1.5 [1.3;1.7] | 1.9 [1.7;2.1] | 0 |
| LVGMT Ogawa,[CI]+ | 1.9 [1.8;2.1] | 2.1 [2;2.3] | 0.1 |
| VSCR Inaba, % [CI]* | 50.8 [37.9;63.7] | 63.5 [51.3;75.7] | 0.2 (0.1) |
| VSCR Ogawa, % [CI]* | 41 [28.3;53.7] | 50.8 [38.1;63.5] | 0.3 (0.2) |
| > 14 years | N = 55 | N = 54 | |
| LVGMT Inaba,[CI]+ | 1.6 [1.4;1.8] | 1.6 [1.4;1.8] | 1 |
| LVGMT Ogawa,[CI]+ | 2 [1.8;2.2] | 1.9 [1.7;2.1] | 0.7 |
| VSCR Inaba, % [CI]* | 50.9 [37.3;64.5] | 46.3 [32.6;60] | 0.60 (0.60) |
| VSCR Ogawa, % [CI]* | 21.8 [10.6;33.1] | 33.3 [20.3;46.3] | 0.2 (0.1) |

+ P-value from Wilcoxon test, * P-value from chi-square, LVGMT = Log of vibriocidal Geometric mean titer, VSCR = vibriocidal seroconversion rate. Adjusted = p.value are from logistic regression

The immune response following administration of a dose of OCV at 9–11 days after administration was higher for serotype Ogawa in each of the age groups of individuals previously exposed to cholera vaccination and was not higher for Inaba. The difference in kinetics of Inaba and Ogawa serotypes may be related to the earlier exposure the Ogawa serotype that was identified during the 2017 outbreak [15–17]. The immune response curve following the first dose of OCV vaccine shows an increasing vibriocidal concentration, peaking at 9–11 days after a single dose OCV administration and then gradually decreasing. The curve following the second dose peaks days 9–11 and days 23–25 but on days 28–30, the immune response among those previously exposed to OCV was not different among those exposed to 1 dose OCV compared to those exposed to two doses. To the best of our knowledge, no study describing the booster immune response curve following the second booster dose has been published, nor has a study reported the booster immune response following the first and second booster doses of OCV in the same population. The data collected in our study do not allow to explain the absence of difference in booster immune response at 28-day following one and two doses of OCV vaccination. This lack of difference may be explained by a local intestinal immune response that is probably triggered by the OCV first dose that partially neutralizes the second dose to reduce its bioavailability and the resulting immune response [18]. The consistency of the observed findings needs to be explored in different settings and over other biomarkers to guide the planning of revaccination populations in need.

The study presented some limitations. The study took place at the start of the COVID-19 pandemic in Cameroon, which led to delays in the supply of vaccines, reducing the enrolment window for eligible participants in the 1–5 age group in 2017. This resulted in the sample size for this age group not being reached. Also, detailed data on the nutritional status of participants that

could have contributed in guiding the understanding of the immune response in the study arms were not collected. Also, the vaccination status of most of the participants was not determined based on vaccination cards as these were unavailable for majority of participants but was determined by tracking using vaccine images and previous period of vaccination.

In summary, the study aimed to verify the hypothesis of a rapid vibriocidal response to participants who had been vaccinated four years earlier. We also intended to determine if the magnitude of the vibriocidal response was higher in those vaccinated earlier. However, we did not observe a rapid vibriocidal response in any of the groups and there was no difference in this response depending on earlier vaccination. Similarly, except for a higher Ogawa in those vaccinated earlier, the vibriocidal titers were similar in the different groups. The study does have implications for determining the number of doses of vaccine to be delivered when revaccinating populations that were vaccinated earlier. Since the previously vaccinated people responded in the same way as those not previously vaccinated, we feel this data suggests that that revaccination of previously vaccinated groups should follow the same recommendations as vaccination of groups not previously vaccinated; that is, two doses is appropriate whenever this is feasible.

## Supporting information

**S1 Checklist. CONSORT 2025 checklist (This checklist is based on the CONSORT 2025 Statement.** Reproduced under the Creative Commons Attribution 4.0 International License (CC BY 4.0)).
(DOCX)

**S1 Data. Study dataset.**
(DTA)

**S1 Appendix. Annex tables for Comparison of Vibriocidal GMT and Seroconversion rates after a single dose of OCV depending on receipt of OCV four years earlier.**
(DOCX)

## Acknowledgments

We acknowledge and thank the participants in Mogode who participated in this research study.

## Author contributions

**Conceptualization:** Jerome Ateudjieu, Ketina Hirma Tchio-Nighie, Collins Buh Nkum, Landry Beyala Bita'a, Sabine Nanfak, Amanda K Debes, David A Sack.

**Data curation:** Jerome Ateudjieu, Etienne Guenou, Carrel Fokou, Sabine Nanfak.

**Formal analysis:** Ketina Hirma Tchio-Nighie, Etienne Guenou, Carrel Fokou, Landry Beyala Bita'a, Sabine Nanfak, David A Sack.

**Funding acquisition:** Jerome Ateudjieu, Amanda K Debes, David A Sack.

**Investigation:** Ketina Hirma Tchio-Nighie, Etienne Guenou, Collins Buh Nkum, Landry Beyala Bita'a, David A Sack.

**Methodology:** Jerome Ateudjieu, Carrel Fokou, David A Sack.

**Project administration:** Jerome Ateudjieu, Ketina Hirma Tchio-Nighie.

**Supervision:** Etienne Guenou, Collins Buh Nkum, Landry Beyala Bita'a, David A Sack.

**Visualization:** Ketina Hirma Tchio-Nighie, Carrel Fokou, Collins Buh Nkum, Winny Dora Ateudjieu-Kenfack.

**Writing – original draft:** Jerome Ateudjieu, Carrel Fokou.

**Writing – review & editing:** Jerome Ateudjieu, Ketina Hirma Tchio-Nighie, Etienne Guenou, Carrel Fokou, Collins Buh Nkum, Landry Beyala Bita'a, Winny Dora Ateudjieu-Kenfack, Sabine Nanfak, David A Sack.

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
