## [Decision Letter · Decision Letter 0]

6 Aug 2025

PGPH-D-25-01336

Serological responses to killed oral cholera vaccine (OCV) when given 4 years after initial receipt of OCV in Cameroon

Dear Dr. Ateudjieu,

Thank you for submitting your manuscript to PLOS Global Public Health. After careful consideration, we feel that it has merit but does not fully meet PLOS Global Public Health’s publication criteria as it currently stands. Therefore, we invite you to submit a revised version of the manuscript that addresses the points raised during the review process.

We look forward to receiving your revised manuscript.

Kind regards,

Espoir Bwenge Malembaka, MD

Academic Editor

Journal Requirements:

1. Please provide additional details regarding participant consent. In the ethics statement in the Methods and online submission information, please ensure that you have specified (1) whether consent was informed and (2) what type you obtained (for instance, written or verbal, and if verbal, how it was documented and witnessed). If your study included minors, state whether you obtained consent from parents or guardians. If the need for consent was waived by the ethics committee, please include this information.

2. We are unable to open your Supporting Information file “Base MOBS treated.dta”. Please kindly revise as necessary and re-upload.

3. Please provide separate figure files in .tif or .eps format.

4. We have amended your Competing Interest statement to comply with journal style. We kindly ask that you double check the statement and let us know if anything is incorrect.

5. Your current Financial Disclosure states, “This work was supported by the Department for International Development and Wellcome Trust: grant number: 215663/Z/19/Z and National Institute of Allergy and Infectious Disease (5R01AI123422)”. However, your funding information on the submission form indicates that you received funding from “Wellcome Trust” with grant number “215663/Z/19/Z”. Please indicate by return email the full and correct funding information for your study and confirm the order in which funding contributions should appear. Please be sure to indicate whether the funders played any role in the study design, data collection and analysis, decision to publish, or preparation of the manuscript.

6. Some material included in your submission may be copyrighted. According to PLOS’s copyright policy, authors who use figures or other material (e.g., graphics, clipart, maps) from another author or copyright holder must demonstrate or obtain permission to publish this material under the Creative Commons Attribution 4.0 International (CC BY 4.0) License used by PLOS journals. Please closely review the details of PLOS’s copyright requirements here: PLOS Licenses and Copyright. If you need to request permissions from a copyright holder, you may use PLOS's Copyright Content Permission form.

Potential Copyright Issues: Figure 1: please (a) provide a direct link to the base layer of the map (i.e., the country or region border shape) and ensure this is also included in the figure legend; and (b) provide a link to the terms of use / license information for the base layer image or shapefile. We cannot publish proprietary or copyrighted maps (e.g. Google Maps, Mapquest) and the terms of use for your map base layer must be compatible with our CC-BY 4.0 license.

Additional Editor Comments (if provided):

In addition to the reviewers' comments, here are some additional observations for the authors:

1. Line 98 : The phrase “randomly stratified by age” appears ambiguous. Stratification and randomization are typically distinct steps in study design. Could the authors clarify whether participants were stratified by age and then randomly selected within each age group, or if a different method was used?

2. In the Methods section of the abstract, the authors mention measuring vibriocidal antibody titers on days 4–6 post-vaccination. However, vibriocidal titers typically peak around day 10 after oral cholera vaccination (see:

https://www.sciencedirect.com/science/article/abs/pii/0264410X94903352 ). Could the authors justify why an earlier time point was chosen and discuss how this may have affected the observed immune response?

3. It is unclear why the vibriocidal titers were measured at different time points post-vaccination in groups A and B. Could the authors provide a clearer rationale for the differences in sampling schedules across groups?

4. The manuscript provides limited information on the logistic regression analyses. Could the authors specify how the linearity assumption was assessed? Additionally, more detail is needed on the covariates included—what specific “household characteristics” were analyzed, and how were these variables selected for inclusion in the model?

5. How many participants had vaccination status confirmed by vaccination cards versus self-report? In many Sub-Saharan African settings where oral vaccines are given regularly, recall bias—especially four years post-vaccination—can be substantial. Would the authors consider performing a sensitivity analysis restricted to participants with card-confirmed vaccination status? This could strengthen the internal validity of the findings and might even be presented as the primary analysis.

6. Could repeated, low-grade or undetected natural exposure to Vibrio cholerae in the study setting have reduced the differences in vibriocidal antibody responses between vaccinated and unvaccinated participants? This potential confounder may warrant discussion in the manuscript.

Reviewers' comments:

Reviewer's Responses to Questions

**Comments to the Author**

1. Does this manuscript meet PLOS Global Public Health’s publication criteria? Is the manuscript technically sound, and do the data support the conclusions? The manuscript must describe methodologically and ethically rigorous research with conclusions that are appropriately drawn based on the data presented.? Is the manuscript technically sound, and do the data support the conclusions? The manuscript must describe methodologically and ethically rigorous research with conclusions that are appropriately drawn based on the data presented.

Reviewer #1: Partly

Reviewer #2: Partly

Reviewer #3: Yes

Reviewer #4: Yes

2. Has the statistical analysis been performed appropriately and rigorously?

Reviewer #1: No

Reviewer #2: Yes

Reviewer #3: Yes

Reviewer #4: Yes

3. Have the authors made all data underlying the findings in their manuscript fully available (please refer to the Data Availability Statement at the start of the manuscript PDF file)?

The PLOS Data policy requires authors to make all data underlying the findings described in their manuscript fully available without restriction, with rare exception. The data should be provided as part of the manuscript or its supporting information, or deposited to a public repository. For example, in addition to summary statistics, the data points behind means, medians and variance measures should be available. If there are restrictions on publicly sharing data—e.g. participant privacy or use of data from a third party—those must be specified.requires authors to make all data underlying the findings described in their manuscript fully available without restriction, with rare exception. The data should be provided as part of the manuscript or its supporting information, or deposited to a public repository. For example, in addition to summary statistics, the data points behind means, medians and variance measures should be available. If there are restrictions on publicly sharing data—e.g. participant privacy or use of data from a third party—those must be specified.

Reviewer #1: Yes

Reviewer #2: Yes

Reviewer #3: Yes

Reviewer #4: Yes

4. Is the manuscript presented in an intelligible fashion and written in standard English?

Reviewer #1: Yes

Reviewer #2: No

Reviewer #3: Yes

Reviewer #4: Yes

Reviewer #1: This is an interesting study, addressing important questions. However the study design suggests that its an RCT, however the title does not reflect this and is therefore misleading and so should be updated.

As this reported as randomised controlled trial - the manuscript shoulf follow CONSORT reporting guidelines, as they are some fundamental methodological elements not reported in the manuscript.

Some comments :

Sample size section- can this be written so that its clear the total number of participants required, its a difference in proportion between control and intervention, and so should be presented to reflect that and also inflated for the drop-out rate. Because earlier there is mention to two stratification factors age and exposure status. In the abstract a number of 350 is given but 430 were approached?

Note that stratification factors are considered to ensure balance between group, i.e control and intervention.

Randomization - how was allocation concealment accounted for. Since the block size is 2 - which really should have been variables, i.e 2, 4 to discourage predictability.

What was the randomisation system used, i.e randomisation schedule used, computer generated and who generated the reandomisation schedule?

What was the process when randomising partcipants - all thes details need to be included in the manuscript?

Any blinding, if so who? if not then state if this open label and the rataionale.

Statistical analysis section requires more information:

- Define population for analysis

- Any indication of how missing data would be handled?

- Primary analysis, with the number of participants enrolled, doing a simple t-test is not best approach as the analysis is not accounting/adjusting for the stratification factors - can authors perform some regression analysis - its mentioned further on in the stats section - but still unclear.

-Same comments for secondary analyses

Results:

Table 1 - as this randomised controlled study - not advisable to test for differences as baseline, as any differences observed would be by chance - so advise to omit p-value column and updated stats section accordingly.

Table 1 - keep use of terms consistent in the report, i.e Arm A, Cohort A ?

Table 1 - Age median(mean) - usually presented as Age - Median (IQR), OR Age - Mean (SD) - certainly present both.

For the variables in table - not expecting to see CI, either range, IQR or SD should be given - please update.

Equally for Table 2, where these comparison pre-specified, suggested to keep as descriptive analyses,

The main comparisons should ideally be Arm A vs Arm B.

The authors should really excerse caution with so many tests - that may not have been necessarity prespecified and little numbers, most results should be limited to descriptive analyses.

Reviewer #2: In this manuscript, the authors aim to compare vibriocidal responses among individuals during a re-vaccination campaign conducted 4 years following a previous campaign in Cameroon.

This is an important and carefully designed study. However, the manuscript requires substantial revision to improve clarity and accuracy. The authors also need to revisit their conclusions in light of the results.

The methods and results sections contain numerous inconsistencies in numbers and study design elements (such as unclear references to what Arms A and B represent). Additionally, much of the methods and results sections are difficult to follow. The manuscript needs to be rewritten with particular attention to clearly describing comparisons and findings for each result, especially given the study's complex design. Below, I have detailed specific instances of these issues, along with broader comments regarding the conclusions that must be addressed before publication.

Main comments:

1) The study was designed to compare age-stratified rapid vibriocidal responses following OCV between individuals who had been vaccinated 4 years earlier and those who had not. The researchers found no significant difference between these groups. However, this finding alone does not support the authors' conclusion that two doses are required for re-vaccination campaigns (abstract lines 46-47). This is particularly true given that they did observe significantly increased vibriocidal responses for single-dose revaccination at day 9, and considering that rapid vibriocidal responses have not been established as a correlate of protection.

2) Does Arm A and B refer to receipt of 1 or 2 doses in the past (as described in the results lines 252-253) or in the present study (as described in the methods and Figure 2)? This is crucial information that must be clarified consistently throughout the manuscript.

3) If Arms A and B correspond to what is described in the methods and Figure 2, are the comparisons between these arms shown in Table 5 and Figure 5? Please clarify this relationship. If so, the results suggest that participants did not exhibit a stronger vibriocidal response following two-dose re-vaccination compared to one-dose re-vaccination. This finding does not support the conclusion stated in the abstract that two doses are needed for re-vaccination campaigns.

Additional comments:

• The full manuscript would benefit from careful copyediting. For example, lines 31-34 in the abstract are difficult to follow, and could be broken up into two sentences.

• The results section in the abstract is also difficult to follow and should be revised. This section should also include confidence intervals and statistical tests conducted.

• Line 35 and line 41 in the abstract. Suggest including the specific time intervals instead of “periodically” and “earlier”, respectively.

• Line 80 in the introduction should clarify that the youngest age group examined is 4-year-olds, not 1–4-year-olds, as children under 4 were excluded from the study according to line 96.

• Line 100. “Samples were collected periodically”. Please include the specific time intervals.

• Line 116 states that individuals 1 and older were included, but this is inconsistent with the study design, where it stated that individuals 4 and older were included.

• In the sample size section, please clarify how sample size calculations considered Arm A and B of the intervention. As written, only Arm A is described.

• In Table 1, please clarify if Cohort A and Cohort B are the same as Arm A and Arm B. Based on Figure 3, is appears and Cohort A and B refer to people who did and did not received the vaccine previously, which is confusing as Arm A and B refer to those that received 1 or 2 doses in this study. This needs to be made very clear in the manuscript.

• All figures require more detailed legends.

• Lines 263-267. These sentences are very general. Please be explicit in what was found.

• What do Arm A and Arm B refer to in Figure 5? Please also include a more detailed description of these results in the text on lines 307-308.

I used Claude.ai to copyedit some sentences of my review to ensure that my meaning was described clearly. I did not copy any content from the manuscript into the AI tool.

Reviewer #3: 1. Although the manuscript notes that the study intervention (OCV) was administered four years after prior vaccination, it would enhance clarity to explicitly state the actual study period (i.e., study start and end dates).

2. In Figure 1, it is unclear whether the first blood draw was conducted prior to vaccination (as a baseline) or after vaccination. Clarifying this would help the reader interpret the immunogenicity results appropriately.

3. It appears that adverse events were reported during the study period. If such data are available, including the conclusion drawn from those safety data, would strengthen the manuscript by providing insight into the safety of administering additional OCV doses in individuals with prior vaccination history.

4. In the baseline characteristics, variables such as "cement/brick/tiles" and "use of improved toilet facilities" may be context-specific. A brief explanation of the relevance of these indicators would help for clarity.

5. In this clinical trial, the procedures involved multiple blood draws and administration of OCV, which may have contributed to protocol deviations. If a per-protocol set (PPS) was pre-defined, it would be informative to include the number of participants in the PPS in the CONSORT diagram, especially if it differs substantially from the full analysis set.

6. It might be helpful to also provide GMT without log-transformation in the table as seen in other OCV publications.

7. The discussion mentions malnutrition as a possible contributor to immune response variability. If heights were collected along with the weight data provided, it would be helpful to include BMI in the demographic table to better support this interpretation.

8. Regarding the study’s conclusions about revaccination with one versus two doses, the following limitations would need to be acknowledged:

a) The sample size seems relatively small, particularly within certain age strata.

b) The study focuses solely on short-term immunogenicity outcomes without evaluating clinical efficacy against cholera, which warrants caution when interpreting the findings in the context of public health policy and vaccine recommendations.

Reviewer #4: Team has written an important manuscript on oral cholera vaccine. Spacing is an important issue that policy makers need to revisit for OCV. Therefore, such type of research should come out quickly to know the picture of OCV. Few comments that need to be discussed in the manuscript.

1. Why authors choose 4 years interval?

2. Is there any differences in the vibriocidal titer in different age group after 4 years vaccination?

3. When OCV production is really challenging to serve the demand even for a single dose, how feasible to think about double dose?

Thank you.

**Do you want your identity to be public for this peer review?** For information about this choice, including consent withdrawal, please see our Privacy Policy..

Reviewer #1: No

Reviewer #2: No

Reviewer #3: No

Reviewer #4: No

---

## [Decision Letter · Decision Letter 1]

18 Jan 2026

PGPH-D-25-01336R1

Serological responses to killed oral cholera vaccine (OCV) when given 4 years after initial receipt of OCV in Cameroon: A randomized controlled trial

Dear Dr. Ateudjieu,

Thank you for submitting your manuscript to PLOS Global Public Health. After careful consideration, we feel that it has merit but does not fully meet PLOS Global Public Health’s publication criteria as it currently stands. Therefore, we invite you to submit a revised version of the manuscript that addresses the points raised during the review process.

We look forward to receiving your revised manuscript.

Kind regards,

Espoir Bwenge Malembaka, MD

Academic Editor

Journal Requirements:

Additional Editor Comments (if provided):

Dear Authors,

Thank you for submitting your revised manuscript. We appreciate the effort invested in responding to the reviewers’ comments. However, after careful evaluation of the revised version and the authors’ responses, we conclude that the manuscript still requires substantial improvement before it can be considered for acceptance. A major revision is therefore required.

In addition to the comments from the reviewers, we outline below the main issues that must be addressed to improve clarity, methodological rigor, and interpretability of the findings.

Abstract

• The background section of the abstract is overly long, in places more detailed than the results section. Please streamline this section.

• The conclusion of the abstract contains French-influenced expressions and requires correction.

• The abstract conclusion should be strengthened. Rather than restating results, the authors should explain what the findings imply in practical and programmatic terms, particularly in relation to the issue of OCV revaccination raised in the background.

• Overall, the manuscript would benefit from thorough proofreading, particularly to improve grammar, punctuation, and phrasing.

Background

The first sentences are too generic and non-informative.

Lines 55-57: The term “high case-fatality rate” is vague. Could the authors provide some data (numbers) to support the claims here? In many endemic settings, cholera case-fatality rates have declined substantially over the past decades due to improvements in rehydration therapy. Please clarify what is meant here and ensure it is supported by the cited references.

The statement regarding the “negative impact on the performance of health systems” is unclear. It is not evident whether the cited references directly assessed health system performance. Please clarify or rephrase.

The paragraph spanning lines 66–89 is overly long and should be split into at least two paragraphs for improved readability.

Lines 90 to 94: the text is presented as a study protocol for a study that will be conducted later. The authors could consider reformulating it.

Methods

1. Ascertainment of vaccination status

• The authors indicate that only 7 participants had vaccination confirmed by card, meaning vaccination status was largely self-reported. It is unclear how OCV was differentiated from other vaccines administered through routine or campaign activities. Given the 4-year interval since vaccination, recall bias and confusion, particularly among children in LMICs exposed to multiple oral vaccines, are important concerns. These issues must be clearly described in the Methods section and acknowledged as limitations.

2. Statistical analysis

• The statistical methods section remains poorly written and inconsistent, despite previous editorial and reviewer comments.

• It is unclear whether seroconversion was analyzed as a binary or continuous variable. The manuscript mentions a z-test, while tables refer to chi-square tests.

• The methods describe Student’s t-test, but table legends refer to Wilcoxon tests.

• These inconsistencies must be resolved, and the statistical approach clearly and consistently described.

• Details about how the logistic regression was done remain patchy. The description in lines 241–242 is insufficient.

Nutritional status

Nutritional status is discussed as an important factor, yet its assessment is poorly described in the methods. Please specify:

• anthropometric measurements collected,

• instruments used,

• indices calculated,

• and classification criteria applied.

• The uniform use of BMI across adults and children is inappropriate. WHO-recommended indices (BMI-for-age for children and adolescents, BMI for adults) should be used. Please revise accordingly.

• If there are data limitations, the authors should explain in the discussion which data were missing.

Household-level variables included in regression models must be explicitly defined in the Methods, including:

• variable definitions,

• scales of measurement,

• and rationale for inclusion (and if possible references)

6. Wealth, sanitation, and household characteristics

• The manuscript includes a limited set of household variables to describe wealth and living conditions, but the rationale for selecting these specific variables is not explained.

• Previous studies have shown that in some settings the type of sanitation infrastructure alone does not necessarily predict cholera risk. Rather, sharing status, which affects maintenance, cleanliness, and hygiene practices and is strongly correlated with poverty, has been shown to be a more relevant determinant of cholera transmission.

• In this context, poverty and socio-economic status cannot be adequately captured by single variables such as sanitation type or housing material alone. Also, what do the authors mean by housing materials? Do they mean materials for the external walls or materials for the floor?

The authors should therefore:

• justify why the selected variables are considered the most relevant and sufficient indicators of household wealth and cholera risk;

• Explain why other routinely used socio-economic indicators (e.g., electricity in the home, shared sanitation facilities, household crowding, etc.) were not included.

• Provide references supporting the validity of their chosen variables in the study setting

• This justification should be clearly presented in the Methods section.

3. Randomization, p-values, and group imbalance

One reviewer suggested removing p-values, noting that participants were randomized to different vaccination regimens.

While we understand this rationale, the presence of apparent imbalances between groups (for example, housing materials in Table 2) raises questions regarding the effectiveness of the randomization. It also raises the question of whether including p-values in some tables would be helpful for the reader to better understand these imbalances.

The authors should therefore clearly explain:

• how they interpret the observed imbalances between groups,

• and whether these imbalances are compatible with chance alone or suggest implementation issues. If imbalances exist, the analytical strategy used to account for them should be clearly described in the methods section.

Results

1. Table presentation

Tables are difficult to interpret, likely due to insufficient methodological clarity.

In Table 1, it should be clearly explained why some variables are summarized using median (IQR) while others use mean with 95% CI. If IQRs are used, please specify the 25th and 75th percentiles.

The choice of summary statistics must be justified in the Methods section.

2. Table 3

Table 3 is confusing, particularly the final column.

The authors state they compare zero doses vs. one or two doses, yet only one column of data is shown. Please clarify whether this column represents descriptive data or a comparison and revise the table accordingly.

3. Tables and legends

Table legends should be substantially improved.

Given that several tables include different types of data within the same table (e.g., descriptive statistics, comparative statistics, regression outputs), the legends must clearly explain what type of data are presented, what comparisons (if any) are being made, and how the data should be interpreted.

Methodological details currently embedded in table legends should be moved to the Methods section, while legends should focus on guiding data interpretation.

4. Effect estimates

The authors predominantly report p-values, rather than 95% confidence intervals. We strongly recommend reporting 95% CIs, which convey both statistical significance and precision and are generally preferred over p-values alone.

Figures

Figure legends should be expanded to allow figures to be interpreted independently.

In particular, the authors should clearly specify what the points represent and what the error bars represent (e.g., standard deviation, standard error, or 95% confidence intervals)

Conceptual consistency

In the Background (lines 86–89), one of the stated research questions concerns whether serum vibriocidal responses differ between children and older age groups.

However, in response to Reviewer #4 (comment #3: “Is there any differences in the vibriocidal titer in different age group after 4 years vaccination?”), the authors state that “This was not part of our data analysis plan.”

This response is difficult to reconcile with the manuscript text. If the data allow—even in a post hoc analysis—this important question should be addressed. Not doing so would represent a missed opportunity.

Missing data: one reviewer asked a question about how the missing data were handled. It is not clear the authors responded to this question.

Discussion

Paragraphs 406–412 largely repeat the results and do not provide interpretation.

Pages 424-425: “In summary, we had hoped to observe a rapid vibriocidal response to participants who had been vaccinated four years earlier.” It seems the authors are expressing a feeling here. Some might see this as a sign that they were less interested in the truth than in an expected result. Could they rephrase this passage in a more neutral and objective way?

These paragraphs should be revised to offer contextual interpretation, implications, and comparison with existing literature, rather than restating findings.

In summary, while the study addresses an important public health question, substantial revisions are required to improve methodological transparency, analytical rigor, and clarity of presentation. We encourage the authors to carefully address each point above and to provide a detailed, point-by-point response with their revised manuscript.

We look forward to receiving a thoroughly revised version.

Sincerely,

Espoir Bwenge Malembaka

Reviewers' comments:

Reviewer's Responses to Questions

**Comments to the Author**

Reviewer #2: (No Response)

Reviewer #3: All comments have been addressed

publication criteria? Is the manuscript technically sound, and do the data support the conclusions? The manuscript must describe methodologically and ethically rigorous research with conclusions that are appropriately drawn based on the data presented.? Is the manuscript technically sound, and do the data support the conclusions? The manuscript must describe methodologically and ethically rigorous research with conclusions that are appropriately drawn based on the data presented.

Reviewer #2: Yes

Reviewer #3: Yes

3. Has the statistical analysis been performed appropriately and rigorously?

Reviewer #2: Yes

Reviewer #3: Yes

4. Have the authors made all data underlying the findings in their manuscript fully available (please refer to the Data Availability Statement at the start of the manuscript PDF file)?

The PLOS Data policy requires authors to make all data underlying the findings described in their manuscript fully available without restriction, with rare exception. The data should be provided as part of the manuscript or its supporting information, or deposited to a public repository. For example, in addition to summary statistics, the data points behind means, medians and variance measures should be available. If there are restrictions on publicly sharing data—e.g. participant privacy or use of data from a third party—those must be specified.requires authors to make all data underlying the findings described in their manuscript fully available without restriction, with rare exception. The data should be provided as part of the manuscript or its supporting information, or deposited to a public repository. For example, in addition to summary statistics, the data points behind means, medians and variance measures should be available. If there are restrictions on publicly sharing data—e.g. participant privacy or use of data from a third party—those must be specified.

Reviewer #2: Yes

Reviewer #3: Yes

5. Is the manuscript presented in an intelligible fashion and written in standard English?

Reviewer #2: Yes

Reviewer #3: Yes

Reviewer #2: The authors have mostly addressed my comments, though there remain errors that I pointed out in my first review that have not yet been addressed:

1) The methods state on line 122 that individuals >1 were included, but on line 103 that individuals >4 were included. Which is correct?

2) Numerical inconsistencies remain. The sample sizes in Table 4 are not consistent with the sample sizes in Table 3. In addition, the sample sizes in Table 5 do not match the sample sizes in Table 1 and are substantially smaller. Are these errors in data entry, or were participants lost to follow up? This needs to be clearly addressed in the paper. If this represents loss to follow up, include a description of how characteristics of the participants with missing data compare to those that remain to address potential biases.

Additional copy-editing is required:

- There are some French words in the abstract line 49.

Reviewer #3: The manuscript shows general improvement through revision; however, minor corrections are still required. Recommend to perform an overall check of the words used, the numbers mentioned, and table formatting. A few minor comments are provided in the attached PDF.

**Do you want your identity to be public for this peer review?** For information about this choice, including consent withdrawal, please see our Privacy Policy..

Reviewer #2: No

Reviewer #3: No

---

## [Editor Report · Decision Letter 2]

30 Mar 2026

Serological responses to killed oral cholera vaccine (OCV) when given 4 years after initial receipt of OCV in Cameroon: A randomized controlled trial

PGPH-D-25-01336R2

Dear MD Ateudjieu,

We are pleased to inform you that your manuscript 'Serological responses to killed oral cholera vaccine (OCV) when given 4 years after initial receipt of OCV in Cameroon: A randomized controlled trial' has been provisionally accepted for publication in PLOS Global Public Health.

Best regards,

Espoir Bwenge Malembaka, MD

Academic Editor